# ConCISE: Consensus Annotation Propagation of Ion Features in Untargeted Tandem Mass Spectrometry Combining Molecular Networking and *In Silico* Metabolite Structure Prediction

**DOI:** 10.3390/metabo12121275

**Published:** 2022-12-16

**Authors:** Zachary A. Quinlan, Irina Koester, Allegra T. Aron, Daniel Petras, Lihini I. Aluwihare, Pieter C. Dorrestein, Craig E. Nelson, Linda Wegley Kelly

**Affiliations:** 1Scripps Institution of Oceanography, University of California San Diego, San Diego, CA 92093, USA; 2Department of Chemistry and Biochemistry, University of Denver, Denver, CO 80210, USA; 3Skaggs School of Pharmacy and Pharmaceutical Sciences, University of California San Diego, San Diego, CA 92093, USA; 4Cluster of Excellence “Controlling Microbes to Fight Infections” (CMFI), University of Tuebingen, 72074 Tuebingen, Germany; 5Center for Microbial Oceanography, Research and Education, Department of Oceanography and Sea Grant College Program, University of Hawai’i at Mānoa, Honolulu, HI 96822, USA

**Keywords:** annotation propagation, CANOPUS, metabolomics, dissolved organic matter

## Abstract

Recent developments in molecular networking have expanded our ability to characterize the metabolome of diverse samples that contain a significant proportion of ion features with no mass spectral match to known compounds. Manual and tool-assisted natural annotation propagation is readily used to classify molecular networks; however, currently no annotation propagation tools leverage consensus confidence strategies enabled by hierarchical chemical ontologies or enable the use of new *in silico* tools without significant modification. Herein we present ConCISE (Consensus Classifications of *In Silico* Elucidations) which is the first tool to fuse molecular networking, spectral library matching and *in silico* class predictions to establish accurate putative classifications for entire subnetworks. By limiting annotation propagation to only structural classes which are identical for the majority of ion features within a subnetwork, ConCISE maintains a true positive rate greater than 95% across all levels of the ChemOnt hierarchical ontology used by the ClassyFire annotation software (superclass, class, subclass). The ConCISE framework expanded the proportion of reliable and consistent ion feature annotation up to 76%, allowing for improved assessment of the chemo-diversity of dissolved organic matter pools from three complex marine metabolomics datasets comprising dominant reef primary producers, five species of the diatom genus *Pseudo-nitzchia,* and stromatolite sediment samples.

## 1. Introduction

High-throughput metabolomics has enabled the screening of hundreds of thousands of metabolites which have the potential to define biological and ecological samples more thoroughly than has previously ever been possible [1]. One of the most promising applications of these emerging tools is understanding the composition and transformation of the hyper-diverse samples such as dissolved organic matter, which comprises a complex mixture of carbon-based molecules ubiquitous in aquatic and terrestrial habitats [2,3]. Characterizing the chemical diversity of these dissolved organic matter (DOM) pools is vital for understanding how changes in metabolite availability can select for certain microbial communities [4]. Untargeted liquid chromatography tandem mass spectrometry (LC-MS/MS) and the development of new techniques to extract molecules from complex samples has enabled high-throughput detection of tens of thousands of molecular features from individual samples across diverse ecosystems [1,5,6,7]. These ecosystem metabolomes can be characterized through spectral matching to reference libraries (e.g., GNPS, massBank, HMDB, ReSpect, NIST14, NIST17) [8,9,10,11] combined with molecular networking which are both available within the Global Natural Products Social Molecular Networking (GNPS) platform [11,12]. Molecular networking expands the interpretability of untargeted metabolomic samples as compositionally similar ion features are networked together using spectra cosine score similarity. In more complex samples, these spectral library matches might identify fewer than 10% of features in the dataset. To facilitate structural identification of unknown metabolites, multiple *in silico* annotation tools have been recently developed, which assign putative annotations across untargeted mass spectrometry experiments (e.g., SIRIUS, ZODIAC, CSI:FingerID, CANOPUS, Metfrag) [13,14,15,16,17]. 

One approach for expanding the number of annotated features is by manually propagating annotations of neighboring features in molecular networks containing database spectral library matches or *in silico* annotations. However, manual annotation propagation is limited by inherent bias and becomes increasingly time consuming with untargeted metabolomic studies, especially complex and diverse datasets such as DOM [2]. There are currently two tools available which automate the propagation of spectral features within molecular networks [6,18]; however, these tools do not allow for implementation of new *in silico* annotation tools such as CANOPUS [17], nor rely upon annotation consensus within molecular subnetworks. To address these limitations, we present ConCISE: Consensus Classifications of *in silico* Elucidations, a new chemoinformatic tool which leverages spectral library database matches and *in silico* class prediction by establishing network specific consensus classifications from chemical ontologies (ChemOnt, [19]). To illustrate the application of ConCISE we utilized CANOPUS *in silico* class predictions to expand putative annotations between 465% and 1244% in three untargeted metabolomics datasets, while also increasing confidence in those annotations through cross-validation within molecular subnetworks and classification hierarchy.

## 2. Materials and Methods

### 2.1. Availability

ConCISE is available on github (github.com/zquinlan/concise; 10.5281/zenodo.7377913; accessed on 29 November 2022) and can be run on a local machine through a command line interface (CLI) or graphical user interface (GUI) by downloading the source files or executable files. Alternatively, the command line interface can also be run on a virtual machine through myBinder [20] which is maintained within the same github repository. 

### 2.2. ConCISE Workflow

ConCISE requires a connection to the internet, even when run locally, to access the Global Natural Product Social Molecular Networking (GNPS) spectral library matches. The ConCISE workflow requires three main inputs: (1) the GNPS task ID of the molecular networking job. (2) The *in silico* summary file which can either be the CANOPUS summary tab separated values (tsv) file exported directly from SIRIUS or a tsv with ion feature annotations from any other in silico tool using a three-level hierarchical ontology. In the latter scenario the feature number, superclass, class, and subclass columns must be named ‘featureNumber’, ‘superclass’, ‘class’, ‘subclass’. (3) The networking info tsv from the clusterinfo_summary subdirectory in the Cytoscape download from GNPS. The networking tsv may be replaced by any tsv which has a column with feature number and subnetwork number with the columns are named ‘cluster index’ and ‘componentindex’, respectively. Example summary files are available from github.com/zquinlan/concise/exampleFiles (accessed on 29 November 2022).

First, all sub-networks with at least one library annotation from GNPS are identified (Figure 1a). The ChemOnt ontology for all ion features with a library annotation that has structural information (e.g., SMILES or InChI [21,22]) within each subnetwork will be compared at each taxonomic level (superclass, class, and subclass). For a consensus annotation to be selected at the superclass level at least 50% of the nodes within that sub-network must have the same superclass annotation. To ensure the authenticity of the narrower classifications levels of class and subclass a stricter cutoff of 70% consensus between nodes is required (Figure 1b). These two thresholds were empirically selected using case study one to ensure both high annotation and true positive rates. These default thresholds may be modified by the user in all iterations of the workflow (CLI, GUI, myBinder). The ConCISE consensus annotation will select the lowest level (most granular) annotation which passes the cutoff criteria for that classification level and all higher classification levels within the respective annotation taxonomy. The *in silico* consensus annotations follow the same schema as that of the spectral library matches. Ion features not linked in subnetworks (so-called “singleton nodes”) are only given the corresponding library annotation and in the absence of any library match do not receive an *in silico* annotation to reduce the occurrence of false positives.

### 2.3. Running ConCISE

Add the library, GNPS Task ID, CANOPUS summary table and Networking Information table to the required fields (Appendix A). Thresholds will default to 50%, 70% and 70% unless modified by the user in the appropriate field. If running locally, there is an optional field to select where the ConCISE summary file will be written. All dependencies are bundled within the GUI; however, running ConCISE from the command line will require the installation of python (https://www.python.org/ accessed on 29 November 2022; ConCISE was written in python3), as well as the following libraries: “requests” (https://pypi.org/project/requests/ accessed on 29 November 2022), “Pandas” [23]; 10.5281/zenodo.6702671, and “NumPy” [24].

### 2.4. ConCISE Export

ConCISE creates a single csv file into the working directory (or selected directory if included) called ConCISEConsensus.csv. This csv file contains a row for every feature within the molecular networking job, a ConCISE consensus annotation with the full chemical hierarchy of the consensus annotation, a score (percent consensus of annotations within the subnetwork), an ontology level at which ConCISE was able to find a consensus annotation, the number of nodes which were used to build the consensus, and the source for which the annotation classification was derived (library or *in silico*). An example export of ConCISE is available from the github repository (github.com/zquinlan/concise/exampleFiles/ accessed on 29 November 2022).

## 3. Results

### 3.1. Manual Validation of ConCISE Consensus Annotations

ConCISE was tested using three publicly available independent datasets using CANOPUS predicted ClassyFire ontologies described below as *Case Studies* (MassIVE IDs: MSV000082083, MSV000081731, MSV000083729). These datasets include exometabolites from the environment, microbial cultures, and sediment metabolite samples with molecular feature counts ranging from 9904 to 21991 (from 3112 to 7293 molecular subnetworks). In total, 1789 individual library spectral matches from the three datasets were evaluated. To estimate the true positive rate of *in silico* consensus annotations, ConCISE was run without any library matches for each of the test datasets. The resultant consensus annotations from each test were then manually verified against the spectral library match compound identities reported from GNPS. ConCISE consensus annotations from spectral library matches had a true positive rate (TPR) of 98.88% ± 1.62% (Figure 2a) while *in silico* consensus annotations had a TPR of 96.1% ± 4.96% across the ClassyFire hierarchies (Superclass, Class, Subclass). Compared to spectral library matches alone without any prior annotation propagation, ConCISE consensus annotations expanded the subnetwork annotation rate by up to 1244% with 58.6%, 56.3%, and 76.5% of ion features receiving a ConCISE consensus annotation in the three datasets, respectively (Figure 2b). 

### 3.2. Case Study 1—Characterization of the Dissolved Organic Matter Pools from Dominant Coral Reef Primary Producers

Coral reefs are regarded as one of the most important ecosystems both ecologically and economically; touted as cradles of biodiversity, producing the most biodiversity per unit area of any marine-environment [25] and representing one the most productive systems in the world [26,27,28,29]. However, many coral reefs are shifting to fleshy algal dominated systems as the microbial community functions shift towards increased microbial loads and copiotrophic feeding strategies [30,31,32,33]. This shift in reef metabolism is hypothesized to be the result of shifts in the composition of the dissolved organic matter exuded into the water by the benthic primary producers of coral reefs [34]. Understanding the composition of the exuded dissolved organic matter is vital to understanding the drivers of coral reef phase shifts. Untargeted metabolomics allows for the analysis of these dissolved organic matter pools; however, the spectral annotation rate remains too low to characterize chemical class variability between coral reef primary producers [1,2,3,7]. 

Using the ConCISE workflow, we were able to annotate 61.09% of all molecular subnetworks produced by coral reef primary producers (Figure 2b). These annotations (spectral library match or *in silico*) had an average subnetwork consensus of 87.4% (Figure 3a). Indeed, even in subnetworks with many annotated features, such as in subnetwork 131 (*n* = 43 annotated ion features), had a consensus of 95.3%. This was true for the narrower subclassifications as well including fatty amides (subnetwork 2263) which still had a 72.0% consensus. Moreover, even at the superclass level differences between fleshy algae (*Dictyota ceylanica* and turfing algae) and coral (*Pocillopora verrucosa* and *Porites lobata*) were apparent (Figure 3b). At the subnetwork level, both species of coral produced relatively higher amounts of organic acids while the turfing algae released relatively high amounts of organoheterocyclic compounds. Deeper analysis into the classes and subclasses of these exudates will greatly expand our understanding of the driving forces behind microbial-driven phase shifts which would not possible without ConCISE.

### 3.3. Case Study 2—Differentiating the Exometabolites of Five Pseudo-nitzschia Species

*Pseudo-nitzschia* is a genus of marine diatoms, which can form harmful algal blooms through the production of domoic acid, a neurotoxin that bioaccumulates in aquatic food webs endangering wildlife and human health. Though present worldwide, the specific conditions which promote *Pseudo-nitzschia* blooms and domoic acid production are poorly understood. Previous studies have shown that microbiomes of different *Pseudo-nitzschia* are species-specific [35,36] and microbial interactions mediated by metabolites can promote *Pseudo-nitzschia* growth [37]. A study by Koester et al. (2022) revealed that metabolomes are also species-specific and that nitrogen-containing compounds differentiated the *Pseudo-nitzschia*-microbiome cultures [38]. The prior study used both spectral matches and CANOPUS probabilities to illuminate the metabolomes of *Pseudo-nitzschia* cultures, however, propagation was limited to manual inspection. Using the ConCISE framework, we were able to demonstrate which compound classes are most specific to each of the five *Pseudo-nitzschia* cultures (Figure 4). The two toxic species of *Pseudo-nitzschia* clustered together when incorporating the relative production of each ConCISE classification, which agrees with the findings on feature level [38]. Within this case study, 56.3% of ion features received a putative consensus annotation with a mean consensus of 87.8%.

### 3.4. Case Study 3—Chemical Surey of Schoenmakerskop in the Eastern Cape of South Africa

Stromatolites, or sediments formed by the metabolism of microbial communities rich in cyanobacteria, exist as fossils but also as modern life forms found across the globe. While the microbial makeup of modern sites has been described, studies characterizing the small molecule metabolites produced in the stromatolite barrage pools are limited. We performed a molecular survey of a stromatolite barrage pool found at Schoenmakerskop in the Eastern Cape of South Africa to better understand the small molecule metabolites present in the pool. Furthermore, we were particularly interested in any bioactive molecules produced by cyanobacteria. This case study received the greatest coverage of putative annotations (76.5%) and highest mean consensus of 89.7%.

## 4. Discussion

Untargeted mass spectrometry has the potential to revolutionize our ability to characterize diverse pools of compounds. Low annotation rates hinder the application of untargeted metabolomic approaches [1,5]. Several tools have been developed to improve annotation rates by assigning putative chemical classifications [15,16], which can then be propagated across molecular subnetworks [6,18]. Machine learning approaches such as CANOPUS are able to predict molecular structures on compound class level de novo at both higher annotation and true positive rates than other tools [17]. Currently no annotation propagation tools allow for incorporation of newly developed structural prediction tools nor contribute to annotation validation. ConCISE was developed with a modular capacity to leverage the advancement of *in silico* annotations, combined with the precision of spectral matches to database libraries, to elevate the proportion of classified chemical features. Moreover, ConCISE incorporates consensus annotation propagation as a method to cross-validate structural predictions in conjunction with chemical hierarchy. Here, we illustrate the potential of ConCISE to leverage spectral matches and CANOPUS annotations to characterize metabolite diversity across molecular networks, identify exometabolites from coral, algae and diatom-microbiome associations as well as fossilized stromatolite pools. ConCISE was able to increase subnetwork putative annotations up to 76% in one of the test datasets (58.6%, 56.3%, and 76.5% in our three case studies), while maintaining an average subnetwork annotation consensus above 87%. This convergence of high annotation rate with feature annotation agreement allows for more thorough analysis of molecular diversity in non-targeted experiments.

One caveat to the manual verification of ConCISE consensus annotations against spectral library matches is that these features represent a subset of spectra which have previously been well characterized and could therefore skew the true positive rate as these spectral libraries are often used in development of *in silico* prediction tools. However, CANOPUS has shown to have very high precision across molecular classifications, outcompeting currently available *in silico* annotation tools including CSI Kernel, CSI:FingerID and MetFrag, alluding to its ability to predict previously unclassified ion features [17]. Currently, ConCISE is limited to chemical ontology for consensus classifications which may not be useful in all circumstances and more precise annotations are needed for ion identification. As more spectra are characterized, and *in silico* structural prediction tools continue to be developed, these annotations will only become more accurate with the potential to allow consensus annotations to be assigned from more narrow annotations. Additionally, the active and open-source development will allow for continual maintenance of ConCISE alongside the development of mass spectrometry techniques and *in silico* tools. 

## 5. Conclusions

ConCISE is a unique annotation propagation tool as it provides cross-validation of *in silico* predicted structures with the derivation of consensus annotations along with a modular design which will be able to incorporate future structural prediction tools. ConCISE is available as an open-source graphical user interface, command line interface and virtual machine-run jupyter notebook. We have illustrated that through expansion of robust annotations across molecular network through the ConCISE framework enhances our understanding of complex metabolite pools, microbial-mediated interactions, and dissolved organic matter cycling. The ability to characterize these complex molecular networks alludes to this tool’s broad applicability across a myriad untargeted metabolomics experiments.

## Figures and Tables

**Figure 1 metabolites-12-01275-f001:**
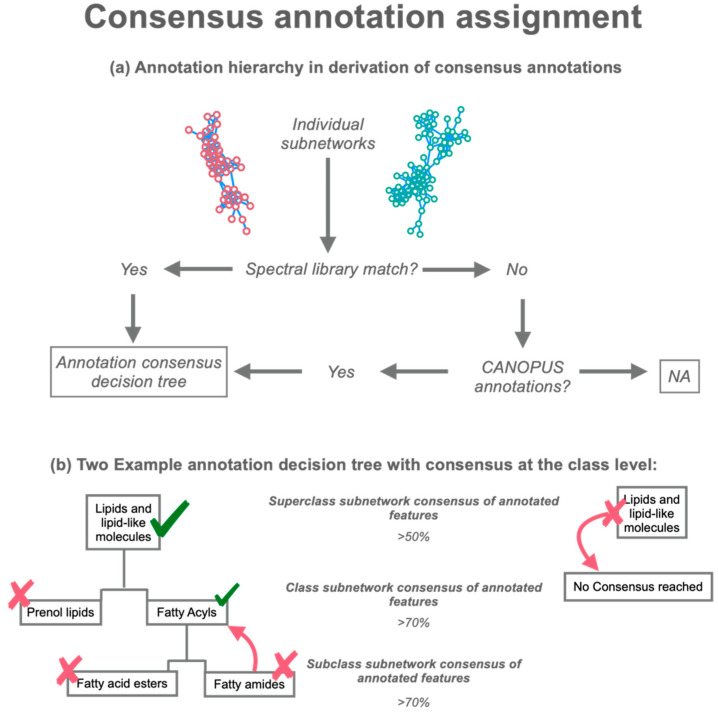
ConCISE consensus annotation pipeline. (**a**) Assignment hierarchy; identification of annotation sources within each molecular subnetwork. Spectral library matches are utilized prior to annotations from *in silico* annotation tools. (**b**) Consensus derivation decision tree used for either library spectral matches or *in silico* matches. Nodes with annotations must have consensus (>50%, >70%, >70%) at each classification level, respectively, (superclass, class and subclass) to receive the higher classification. If the nodes within the subnetworks cannot find a consensus above 50% at the superclass level, then ConCISE returns “No Consensus Reached” for that subnetwork.

**Figure 2 metabolites-12-01275-f002:**
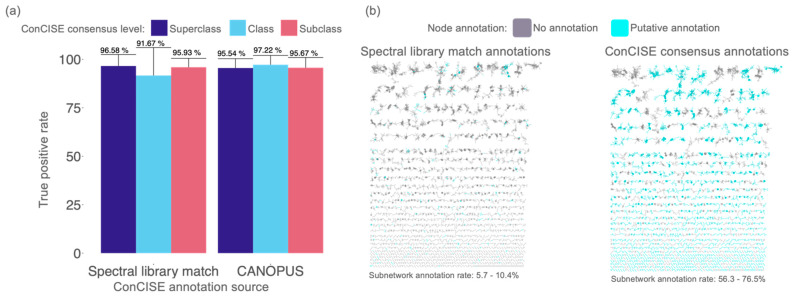
Manually verified true positive rate of ConCISE consensus annotations and increase in annotation rate. (**a**) True positive rate of each classification level in all three validation datasets. (**b**) Example network (MassIVE ID: MSV000082083) illustrating annotation rates from spectral library matches (**left**) and ConCISE (**right**) with putative annotations colored in light blue. Ranges in annotation rate were given for the three datasets used to verify ConCISE. Network annotation rates calculated without single-loop nodes.

**Figure 3 metabolites-12-01275-f003:**
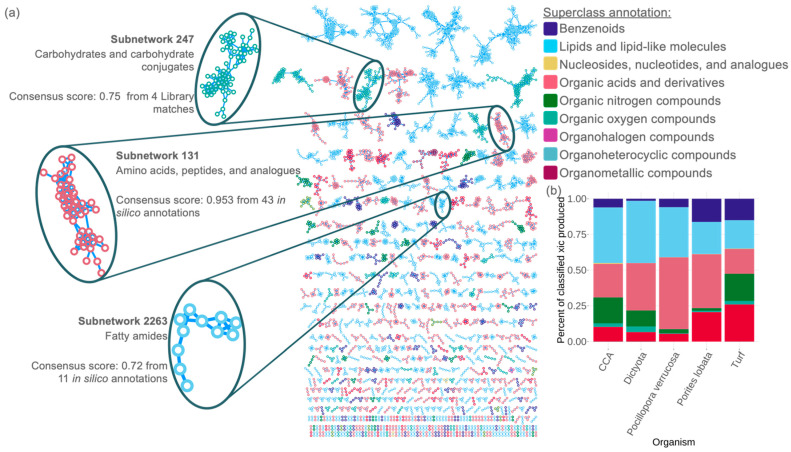
Putative identification of exometabolites released by coral reef primary producers. Dissolved organic matter chemo-diversity from flow-through organism incubations in Mo’orea, French Polynesia of crustose coralline algae (CCA), *Dictyota ceylanica*, turf algae, *Pocillopora verrucosa,* and *Porites lobata*. (**a**) The molecular network of ConCISE classified subnetworks from the incubations with nodes colored by superclass annotation. Three representative subnetworks have been highlighted to illustrate the narrower chemical classification which can be derived from ConCISE annotations along with the consensus score for the annotation within the molecular network. (**b**) Stacked bar chart illustrating the percent of classified extracted ion chromatogram (xic) which falls into each superclassification from each organismal incubation.

**Figure 4 metabolites-12-01275-f004:**
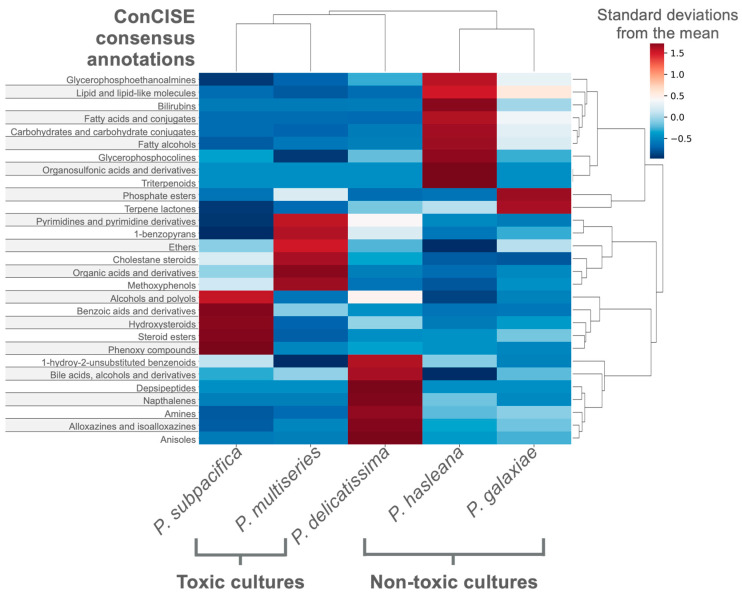
Species-specific exometabolite production from five *Psuedo-nitzchia* species. Cluster dendrogram of total production of the ConCISE classifications which were most variable across the species within the experiment (>1.5 standard deviations from the mean in one treatment). Average classification production for each species was standardized across treatments using a z-score and represented in the color gradient.

## Data Availability

The source code and all data from the case studies used are available on github (github.com/zquinlan/concise) and have been permanently cached at the doi: 10.5281/zenodo.7044828. The original spectra from the case studies are available from massive.ucsd.edu under the ascession numbers: MSV000082083, MSV000081731, MSV000083729.

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
