# Peer review of "ConCISE: Consensus Annotation Propagation of Ion Features in Untargeted Tandem Mass Spectrometry Combining Molecular Networking and In Silico Metabolite Structure Prediction"

_metabolites, 2022, doi:10.3390/metabo12121275_

Round 1
Reviewer 1 Report
The manuscript by Zachary Quinlan et al. described a tool to enhance the annotation of unknowns into class/super-class level by fusing molecular networking, spectral library matching and in silico class predictions to establish accurate putative classifications.
In general, this tool could achieve the aim authors have claimed in the manuscript. It is not surprising that using the subnetwork-based putative annotation to annotate much more unknown ion features by projecting into higher chemical class, or super class. However, molecules are so chemically/biologically different from each other in many cases, even though they are from the same superclass. Chemical class or even subclass is not quite useful in most cases, unless more details (like the potential candidates of the ions) provided for further identification. Authors must clearly discuss this limitation.
More specific comments are given below:
1. Author mentioned at least 50% of the nodes within the subnetwork must have same superclass annotation (line 98). How did you choose 50% for this step? Same question for the following 70% for class and sub-class level (line 100). Can you explain why and make these two parameters user-defined? 2. Line 88, “a tsv with ion feature ChemOnt” is not clear. I understand the tsv file must include ChemOnt indexes for the ion features. Please clarify. Author should provide at least one or two examples/tutorials for further validation and learning purpose. 3. “An example export of ConCISE is available from the github repository” (line 135). Can you add a link here for the example. It is not easy to find the example from a number of files in the github repo. 4. The testing results are performed based on three cases studies from the real world. The composition of these samples are quite complicated. Despite GNPS could provide putative annotation results on the chemical identity. It is inappropriate to use all putative results as the ground truth for benchmarking. In addition, have you ever considered or compared the performance to the neutral loss-based annotation? 5. “Documentation -- Coming soon” from the github repo should be fixed ASAP. Otherwise, there is no way for further using/testing. Author also mentioned, “myBinder” (line 83). Please provide more information on this. In addition, the explanation on the workflow should focus on the general design and mechanism from back-end, it is highly suggested to add the data table formatting details (line 90) etc into the documentation section. How does the propagation network work should be explained here with details. Other comments: Line 86: “the GNPS a task ID” should be corrected as “A GNPS task ID”.Author Response
We appreciate reviewer 1’s comments and believe their comments and recommended changes have improved our manuscript. We agree with all of the revisions made by this reviewer and have addressed each comment below:
In general, this tool could achieve the aim authors have claimed in the manuscript. It is not surprising that using the subnetwork-based putative annotation to annotate much more unknown ion features by projecting into higher chemical class, or super class. However, molecules are so chemically/biologically different from each other in many cases, even though they are from the same superclass. Chemical class or even subclass is not quite useful in most cases, unless more details (like the potential candidates of the ions) provided for further identification. Authors must clearly discuss this limitation.
We have added the below to line 276: “Currently, ConCISE is limited to chemical ontology for consensus classifications which may not be useful in all circumstances and more precise annotations are needed for ion identification. As more spectra are characterized, and in silico structural prediction tools continue to be developed, these annotations will only become more accurate with the potential to allow consensus annotations to be assigned from more narrow annotations.”
We have addressed the reviewers’ other comments bellow:
- Author mentioned at least 50% of the nodes within the subnetwork must have same superclass annotation (line 98). How did you choose 50% for this step? Same question for the following 70% for class and sub-class level (line 100). Can you explain why and make these two parameters user-defined?
The manuscript now reads:
Line 103: “These two thresholds were empirically selected using case-study one to ensure both high annotation and true positive rates. These default thresholds may be modified by the user in all iterations of the workflow (CLI, GUI, myBinder).”
Line 126: “Thresholds will default to 50%, 70% and 70% unless modified by the user in the appropriate field.”
- Line 88, “a tsv with ion feature ChemOnt” is not clear. I understand the tsv file must include ChemOnt indexes for the ion features. Please clarify. Author should provide at least one or two examples/tutorials for further validation and learning purpose.
We have made a new subdirectory in the Github repository zquinlan/Concise/exampleFiles/ which contains two examples of the SIRIUS output (the previous and most recent versions of the SIRIUS export files and the node info file). The manuscript now reads:
Line 95: “Example summary files are available from github.com/zquinlan/concise/exampleFiles.”
- “An example export of ConCISE is available from the github repository” (line 135). Can you add a link here for the example. It is not easy to find the example from a number of files in the github repo.
This file has been added to the exampleFiles subdirectory and a link provided at line 145 which now reads:
“An example export of ConCISE is available from the github repository (github.com/zquinlan/concise/exampleFiles/).”
- The testing results are performed based on three cases studies from the real world. The composition of these samples are quite complicated. Despite GNPS could provide putative annotation results on the chemical identity. It is inappropriate to use all putative results as the ground truth for benchmarking. In addition, have you ever considered or compared the performance to the neutral loss-based annotation?
We had not considered using neutral loss-based annotations because as of right now GNPS only uses modified cosine score clustering. This decision was partially relying upon the research by Bittremieux et al. (2022; Comparison of Cosine, Modified Cosine, and Neutral Loss Based Spectrum Alignment For Discovery of Structurally Related Molecules) which showed better performance of modified cosine score clustering. However, if neutral loss-based clustering was added as an option to GNPS it could be utilized. In theory neutral loss-based clustering could be incorporated into ConCISE already if an analogous node-info dataframe was supplied. We are open to adding additional support to ConCISE in future updates for this and other clustering metrics.
- “Documentation -- Coming soon” from the github repo should be fixed ASAP. Otherwise, there is no way for further using/testing.
We have added more information to the README to assist the general user in downloading and using ConCISE. We have also created a _pages/ subdirectory in the gh-pages branch with documentation for python code. Moreover, we have created a github pages documentation website.
- Author also mentioned, “myBinder” (line 83). Please provide more information on this. In addition, the explanation on the workflow should focus on the general design and mechanism from back-end, it is highly suggested to add the data table formatting details (line 90) etc into the documentation section. How does the propagation network work should be explained here with details.
Line 84 now reads:
“The workflow can be run locally through the GUI, CLI or on a free jupyter notebook virtual machine available through myBinder (linked provided in the github repository readme)”
To make more clear how the propagation network is built we have separated the description into its own paragraph between lines 98 and 114:
“First, all sub-networks with at least one library annotation from GNPS are identified (Figure 1a). The ChemOnt ontology for all ion features with a library annotation that has structural information (e.g. SMILES or InChI [21, 22]) within each subnetwork will be compared at each taxonomic level (superclass, class, and subclass). For a consensus annotation to be selected at the superclass level at least 50% of the nodes within that sub-network must have the same superclass annotation. To ensure the authenticity of the narrower classifications levels of class and subclass a stricter cutoff of 70% consensus between nodes is required (Figure 1b). These two thresholds were empirically selected using case study one to ensure both high annotation and true positive rates. These default thresholds may be modified by the user in all iterations of the workflow (CLI, GUI, myBinder). The ConCISE consensus annotation will select the lowest level (most granular) annotation which passes the cutoff criteria for that classification level and all higher classification levels within the respective annotation taxonomy. The in silico consensus annotations follow the same schema to that of the spectral library matches. Ion features not linked in subnetworks (so-called "singleton nodes") are only given the corresponding library annotation and in the absence of any library match do not receive an in-silico annotation to reduce the occurrence of false positives.”
Other comments: Line 86: “the GNPS a task ID” should be corrected as “A GNPS task ID”.
Now reads: “the GNPS task ID of the molecular networking job”

Reviewer 2 Report
Zachary Quinlan et al for the first time reported that ConCISE (Consensus Classifications of In Silico Elucidations) was used to fuse molecular networking, spectral library matching and in silico class predictions. The ConCISE framework expanded the proportion of reliable and consistent ion feature annotation up to 76%. The findings are interesting, and the manuscript is well-written and structured. There are minor comments for this MS.
Materials and Methods
line 82. the abbreviation should be used in this sentence as it appeared before (graphical user interface (GUI) or command line interface (CLI) ).
Results
Line 208. It would be good to cite the reference here instead of name and year.
Author Response
As suggested by Reviewer 2, we have modified the following:
line 82 now reads: "The workflow can be run locally through the GUI, CLI or on a virtual machine available through myBinder"
Line 208 now includes the citation for Koester et al., 2022.
Round 2
Reviewer 1 Report
The authors have addressed my concerns. I have no more comments